# Coexistence of Bloch and Parametric Mechanisms of High-Frequency Gain in Doped Superlattices

**DOI:** 10.3390/nano13131993

**Published:** 2023-07-01

**Authors:** Vladislovas Čižas, Natalia Alexeeva, Kirill N. Alekseev, Gintaras Valušis

**Affiliations:** 1Department of Optoelectronics, Center for Physical Sciences and Technology (FTMC), Saulėtekio Ave. 3, LT-10257 Vilnius, Lithuania; vladislovas.cizas@ftmc.lt (V.Č.);; 2Institute of Photonics and Nanotechnology, Department of Physics, Vilnius University, Saulėtekio Ave. 3, LT-10257 Vilnius, Lithuania

**Keywords:** superlattice, amplification, large signal, microwaves, terahertz, sub-harmonic, frequency dividers

## Abstract

The detailed theoretical study of high-frequency signal gain, when a probe microwave signal is comparable to the AC pump electric field in a semiconductor superlattice, is presented. We identified conditions under which a doped superlattice biased by both DC and AC fields can generate or amplify high-frequency radiation composed of harmonics, half-harmonics, and fractional harmonics. Physical mechanisms behind the effects are discussed. It is revealed that in a general case, the amplification mechanism in superlattices is determined by the coexistence of both the phase-independent Bloch and phase-dependent parametric gain mechanisms. The interplay and contribution of these gain mechanisms can be adjusted by the sweeping AC pump strength and leveraging a proper phase between the pump and strong probe electric fields. Notably, a transition from the Bloch gain to the parametric gain, often naturally occurring as the amplitude of the amplified signal field grows, can facilitate an effective method of fractional harmonic generation in DC–AC-driven superlattices. The study also uncovers that the pure parametric generation of the fractional harmonics can be initiated via their ignition by switching the DC pump electric field. The findings open a promising avenue for the advancement of new miniature GHz–THz frequency generators, amplifiers, and dividers operating at room temperature.

## 1. Introduction

Semiconductor superlattices, structures formed by alternating two different nanomaterials forming electronic minibands [1], allow the design of carrier transport properties in a desirable way. This feature makes them an attractive media for the exploration of different physical phenomena. In particular, high-electric-field effects in superlattices [2] have gained interest, enabling the possibility to resolve features of ballistic or coherent carrier transport, unveiling the quantum-mechanical nature of excited electronic or excitonic wave packets. Superlattices can exhibit a large variety of intriguing physical effects such as Bloch oscillations [3,4,5], Zener tunnelling [6], coherent Hall effects [7] and superluminal Doppler phenomena [8]. Moiré superlattices in van der Waals heterostructures [9] have enriched the tapestry of these compelling effects due to the unique interplay between atomic structure and electron correlations via pronounced unconventional superconductivity [10], topological edge states [11], correlated insulator phases [12], and high-order fractal quantum oscillations in graphene/h-BN superlattices [13]. On the other hand, semiconductor superlattices can be an essential ingredient in modern electronic and optoelectronic devices; for instance, carrier injectors in infrared and terahertz (THz) quantum cascade lasers [14], amplifiers of electromagnetic radiation, including scattering-assisted inversionless Bloch gain [15,16,17], and sub-terahertz parametric oscillators [18,19]. The latter effect attracts particular attention as it can illuminate the route to tackle the problem of the current lack of compact effective room temperature sources of THz radiation [20]. The superlattice parametric oscillators [18], also known as superlattice multipliers [21,22], are devices based on heavily doped GaAs/AlGaAs superlattices that provide very effective frequency multiplication of a microwave pump field [23], but do not produce a sub-harmonic output. It is also known that with variation in DC bias, the superlattice multiplier can switch to the operational mode of the high-field domain propagation, supporting the generation of high-frequency radiation due to the Gunn effect [24].

Phase-dependent parametric gain, a well-established physical effect, has gained increasing interest due to the growing range of promising applications [25,26,27] and recent experimental demonstrations of the dissipative parametric gain of microwaves in GaAs/AlGaAs superlattices [28]. In detail, by applying a proper combination of DC and AC pump electric fields to a miniband superlattice, the parametric generation of harmonics, half-harmonics (divide-by-2 subharmonics), and various fractional harmonics (divide-by-n sub-harmonics) was observed in a waveguide-based experiment at room temperature [28]. Dissipative parametric generation is caused by a periodic variation in electron mobility rather than a variation in any reactive element. The multilayer nanostructures used in this study were grown by a molecular beam epitaxy on n-doped ($10^{18}$ cm${}^{-3}$) GaAs substrate with an orientation of (100). The superlattice had 30 spatial periods each comprising a 5 nm GaAs quantum well doped with Si and 1 nm Al${}_{0.3}$Ga${}_{0.7}$As as a barrier. This heterostructure design is typical in realising vertical electron transport in miniband superlattices. The active region of the superlattice had a modest doping of $N\approx 10^{16}$ cm${}^{-3}$ fulfilling the Kroemer stability criterion—NL-criterion where *L* denotes the length of the sample—aiming to avoid the formation of high-field domains [29,30].

These experimental findings were shown to be generally consistent [28] with the predictions of the semi-classical model [31,32,33,34,35,36] describing the conditions for the parametric generation of microwaves in DC–AC-driven superlattices. However, these earlier theoretical considerations have a couple of drawbacks that limit their applicability to an explanation of the experimental findings. First, the majority of earlier works employed the small-signal approximation in calculations of the parametric gain with the notable exception of [37], which is exclusively devoted to a rather specific case of the parametric gain in perfectly symmetrical superlattices without DC bias. Second, earlier considerations typically overlooked the phase-independent effects of DC–AC–Bloch gain [38,39] and their interplay with the parametric gain. Importantly, in the framework of the small-signal approximation, the parametric generation of half-harmonics is allowed, but the generation of fractional harmonics, such as divide-by-3 sub-harmonics, is forbidden [31,34]. In [28], the observed excitation of the fractional frequencies was formally attributed to large-signal effects but the physical mechanism behind the phenomena was not elucidated.

In this work, we expand the boundaries of a previous investigation [28] by evaluating the small-signal gain conditions, focusing mainly on the large-signal gain effects when the signal field can be compared with the pump electric field and the Esaki-Tsu critical field. Typically, small-signal gain approximation is used to determine the conditions to initiate generation, while the large-signal model describes the gain limits and maximum power an amplifier or oscillator can deliver. We reveal that in the general case, the amplification mechanism in doped miniband superlattices comprises the coexistence of Bloch and parametric gain, where the contribution of the latter increases with the increase in the AC pump electric field strength. The importance of a proper phase selection and combination of DC and AC pump fields for the manifestation of each mechanism is considered. We also discuss in detail the conditions for the occurrence of all types of sub-harmonics. By combining the emerging understanding of large-signal and Bloch–parametric gain effects in superlattices, we offer a qualitative explanation for the unexpected generation of fractional harmonics observed in the experiment.

## 2. Main Equations and Explanations

We consider a single miniband semiconductor superlattice driven by a combination of constant and coherent alternating electric fields, which are applied along the superlattice axis. Suppose that the total electric field $E_{\mathrm{tot}}\left(t\right)=E_{\mathrm{pump}}\left(t\right)+E_{\mathrm{probe}}$, acting on miniband electrons, consists of the strong DC–AC pump field
(1)$$E_{\mathrm{pump}}\left(t\right)=E_{\mathrm{dc}}+E_{\mathrm{ac}}\cos\left(\omega_{0}t\right)$$
and the probe (or signal) AC electric field
(2)$$E_{\mathrm{probe}}\left(t\right)=E_{1}\cos\left(\omega_{1}t+\phi \right)\;\mathrm{with}\;\omega_{1}=n_{0}\omega_{0}/2\;\mathrm{or}\;\omega_{1}=\left(m/n\right)\omega_{0},$$
where $n_{0}$ is an odd integer (half-integer harmonics), and *n* and *m* are positive integers with no common divisor (n>2, fractional harmonics). Thus, the fields (Equations (Equation 1) and (Equation 2)) have commensurate frequencies and a well-defined phase difference ϕ, essential for parametric amplification and generation. However, if $E_{\mathrm{ac}}=0$ the phase dependence in Equation (Equation 2) can be ignored, and this field configuration is typically used in the analysis of canonical Bloch gain in superlattices [15,40]. The probe field $E_{\mathrm{probe}}$, depending upon the experimental arrangements, can represent either an electromagnetic mode of the external high-Q cavity [31] or an intrinsic longitudinal mode propagating inside the superlattice [28]. We further assume that all fields involved are not too strong in comparison with the miniband width, allowing us to use the standard semi-classical approach to describe the electron transport along the superlattice axis [41]. This semi-classical approach to DC–AC-driven superlattices relies on the solution of the one-dimensional Boltzmann transport equation [41], in some cases with additional calculations based on the Green’s function method [42,43], and also the use of the superlattice balance equations [37,44]. The listed semi-classical methods to solve the Boltzmann equation are rather universal but often computational time-consuming, especially in the case of several parameters vary simultaneously and large-signal effects are involved. Here, we use more simple but fast and reliable methods based on quasi-static approximation.

In the quasi-static approximation [45], the component of the electron drift velocity along the superlattice axis *v* depends on the applied DC–AC electric field $E_{\mathrm{tot}}\left(t\right)$ as
(3)$$v\left(E_{\mathrm{tot}}\right)=2v_{p}\frac{\left(E_{\mathrm{tot}}/E_{\mathrm{cr}}\right)}{\left[1+{\left(E_{\mathrm{tot}}/E_{\mathrm{cr}}\right)}^{2}\right]},$$
where $v_{p}=\Delta d/4\hbar$ is the peak electron velocity (Δ is the miniband width and *d* is the period of the superlattice) and $E_{\mathrm{cr}}=\hbar /ed\tau$ is the critical electric field, and τ is the inter-miniband relaxation time [1]. Since the quasi-static approximation necessitates $\omega_{0,1}\tau \ll 1$ with τ being typically several hundreds of femtoseconds at room temperature [18,46], the response Equation (Equation 3) is valid in a broad frequency range covering both the microwave and low sub-THz frequency domains (≲ 300 GHz).

We assume that the probe electric field can reach strengths comparable to the pump electric field, thus leading to the need for large-signal gain analysis. In simple terms, the small-signal gain model can be used to describe the conditions to initiate generation and the large-signal approach is used to determine the gain limits that can be achieved. To understand whether the probe electric field of a given amplitude can be amplified one needs to calculate the power density
(4)$$P=eN{\langle v\left[E_{\mathrm{tot}}\left(t\right)\right]E_{\mathrm{probe}}\left(t\right)\rangle }_{t},$$
where *P* is the power per active superlattice volume, *N* denotes the electron volume density, and ${\langle \ldots \rangle }_{t}$ denotes the averages over time. Since $E_{\mathrm{tot}}\left(t\right)$ is a periodic field, the averaging can be performed over a common period of the probe and pump AC fields $T_{\mathrm{com}}$ as ${\langle \ldots \rangle }_{t}=\frac{1}{T_{\mathrm{com}}}\int_{0}^{T_{\mathrm{com}}}dt$.

If P<0, the power flows out of the superlattice into an external circuit or cavity, indicating that the device is active and capable of amplifying the probe electric field [47]. For the given pump electric field strength, the probe field can grow further until *P* becomes positive at a certain amplitude of the probe $E_{1}^{\left(s\right)}$. This amplitude corresponds to steady-state oscillations of the superlattice device (in the case of the lossless cavity), a behaviour known as the saturation of gain. The process of generation (stimulated emission) means that P<0 already for $E_{1}\to 0$ and that *P* remains negative up to $E_{1}^{\left(s\right)}$. Thus, for this generation a positive gain exists for both small and large signals when the probe field grows just from small fluctuations.

For every fixed value of the pump ($E_{\mathrm{dc}},E_{\mathrm{ac}}$) and probe $E_{1}$, the power *P* depends on the relative phase ϕ, and the negative power reaches its minimum value at an optimal phase $\phi_{\mathrm{opt}}$ as
(5)$$P\left(\phi_{\mathrm{opt}};E_{1}\right)=\min_{\phi }\left\{P\left(\phi ;E_{1}\right)|E_{1}-\mathrm{fixed}\right\}\;\mathrm{for}\;P<0.$$

It is convenient to introduce dimensionless variables, in which Equation (Equation 4) takes the explicit form
(6)$$\bar{P}=\frac{F_{1}}{2\pi n}\int_{0}^{2\pi n}\frac{2F_{\mathrm{tot}}\left(x\right)dx}{1+F_{\mathrm{tot}}^{2}\left(x\right)},\;\;F_{\mathrm{tot}}\left(x\right)=F_{\mathrm{dc}}+F_{\mathrm{ac}}\cos\left(x\right)+F_{1}\cos\left(\frac{m}{n}x+\phi \right),$$
where the electric fields are represented in units of the critical field $F_{i}=E_{i}/E_{\mathrm{cr}}$, and $\bar{P}=P/P_{0}$ with $P_{0}=eNv_{p}E_{\mathrm{cr}}$. In what follows we will often use these scaled variables to represent our numerical findings. The characteristic power density $P_{0}$ indicates the scale of power that can be generated or amplified in the superlattice ($P≲P_{0}$). It easy to see that the characteristic value of the power density per one electron
(7)$$P_{0}/N=ev_{p}E_{\mathrm{cr}}=\frac{\Delta }{4\tau }$$
depends only on the miniband width. For the wide miniband superlattice used in the experiment [28] Δ=104 meV, $P_{0}/N\approx 21$ nW and therefore the characteristic power generated by $N=10^{16}$ cm${}^{-3}$ electrons homogeneously distributed within the volume *V* (defined by the superlattice mesa dimensions: 80×80μm${}^{2}$ and 180 nm) is $P_{0}V=240$ mW. Everywhere in our estimates we will use this numerical value of $P_{0}V$ and $E_{\mathrm{cr}}\approx 5.5$ kV/cm.

One should note that in addition to the degenerate parametric processes with the frequency relations (Equation (Equation 2)), non-degenerate parametric amplification and generation also exist, where the probe electric field comprises two modes with distinct frequencies satisfying the relations $\omega_{1}\pm \omega_{2}=n_{\pm }\omega_{0}$[28,33,35]. Hereby, $n_{+}$ and $n_{-}$ are integers indicating the number of photons pump participating in the non-degenerate processes. However, within the framework of the quasi-static approximation, the consideration of large-signal effects in non-degenerate parametric processes can be reduced to the corresponding analysis of two selected degenerate processes (see Appendix A for further explanations). This observation allows us to focus on the analysis of the degenerate processes, beginning with an overview of the gain effects related to the small-signal limit of Equation (Equation 4).

## 3. Results and Discussion

### 3.1. Overview of the Small-Signal Results and Introduction to the Large-Signal Effects

We introduce a high-frequency mobility μ to determine the linear response of an electron drift velocity to a weak probe electric field, $v\left(t\right)=\mu E_{1}\cos\left(\omega_{1}t+\phi \right)$. In this small-signal limit ($E_{1}\ll E_{\mathrm{cr}}$), the power is related to the mobility following the well-known expression $P=0.5eN\mu E_{1}^{2}$; hence, the positive small-signal gain P<0 corresponds to μ<0. By using a standard approach [48], the small-signal mobility μ can be represented as the sum of the phase-independent ($\mu_{\mathrm{inc}}$) and phase-dependent ($\mu_{\mathrm{coh}}\left(\phi \right)$) mobility components [28,33,49]
(8)$$\mu \left(\phi \right)=\mu_{\mathrm{inc}}\left(E_{\mathrm{dc}},E_{\mathrm{ac}}\right)+\mu_{\mathrm{coh}}\left(E_{\mathrm{dc}},E_{\mathrm{ac}},\omega_{1}/\omega_{0},\phi \right),$$
(9)$$\mu_{\mathrm{inc}}={\langle v^{\prime }\left[E_{\mathrm{pump}}\left(t\right)\right]\rangle }_{T_{0}},\;\mu_{\mathrm{coh}}={\langle v^{\prime }\left[E_{\mathrm{pump}}\left(t\right)\right]\cos\left(2\omega_{1}t+2\phi \right)\rangle }_{T_{\mathrm{com}}},$$
(10)$$v^{\prime }\left[E_{\mathrm{pump}}\left(t\right)\right]\equiv \frac{\partial v}{\partial E}{|}_{E=E_{\mathrm{pump}}\left(t\right)}.$$

Since the calculation of $\mu_{\mathrm{coh}}$ in Equation (Equation 9) requires averaging over the common period $T_{\mathrm{com}}$, this mobility component demonstrates its dependence on the ratio of the pump and probe frequencies $\omega_{1}/\omega_{0}$. In contrast, the calculation of $\mu_{\mathrm{inc}}$ involves averaging only over the pump electric field period $T_{0}=2\pi /\omega_{0}$; therefore, it depends solely on the pump strength parameters ($E_{\mathrm{dc}},E_{\mathrm{ac}}$). Either $\mu_{\mathrm{coh}}$ or $\mu_{\mathrm{inc}}$, or both, can become negative. These small-signal gain components were coined the coherent gain and incoherent gain, respectively, due to the difference in their dependence on the relative phase ϕ [31,49].

The phase-dependent coherent gain has a parametric nature caused by periodic variation in the differential velocity $v^{\prime }$ under the influence of the AC pump field (Equation (Equation 10)), during which $v^{\prime }$ is able to take negative values [28]. As for other types of parametric processes, the power, associated with the probe field, mainly originates from the alternating pump, while the applied DC bias provides an additional control over the power transfer. However, it is worth noting that in the important case of half-integer harmonics $\omega_{1}=n_{0}\omega_{0}/2$, the coherent mobility $\mu_{\mathrm{coh}}$ becomes zero in an unbiased superlattice ($E_{\mathrm{dc}}=0$) [34]. In this case, achieving positive coherent gain always requires the application of some DC bias that breaks the symmetry of the system [31]. Importantly, the coherent gain always has its maximum at some optimal phase $\phi_{\mathrm{opt}}$. For example, when the probe electric field frequency is second ($\omega_{1}=2\omega_{0}$) or even the multiplier of the pump frequency and $E_{\mathrm{dc}}=0$, $\phi_{\mathrm{opt}}$ is equal to π/2modπ [28]. In this case, the parametric gain can be achieved even in an unbiased superlattice [37].

To identify the physical meaning of incoherent gain, we consider two limiting cases. Firstly, in the the absence of the AC pump electric field, the incoherent mobility (Equation (Equation 9)) takes the form
(11)$$\mu_{\mathrm{inc}}\left(E_{\mathrm{dc}},E_{\mathrm{ac}}=0\right)=\mu_{0}\frac{\left(1-F_{\mathrm{dc}}^{2}\right)}{{\left(1+F_{\mathrm{dc}}^{2}\right)}^{2}},\;F_{\mathrm{dc}}=E_{\mathrm{dc}}/E_{\mathrm{cr}}$$
where $\mu_{0}=2v_{p}/E_{\mathrm{cr}}=e\tau /m_{\mathrm{eff}}$ is the Drude mobility of electrons in the superlattice, and $m_{\mathrm{eff}}=2\hbar^{2}/\Delta d^{2}$ is the effective electron mass at the bottom of the miniband. Gain $\mu_{\mathrm{inc}}<0$ arises when the DC bias exceeds the Esaki-Tsu critical electric field ($F_{\mathrm{dc}}>1$), and then Equation (Equation 11) describes the quasi-static limit of the canonica; Bloch gain (cf. Equation (Equation 11) with the corresponding Equation (Equation 11) in [15]). Secondly, when no DC bias is applied ($E_{\mathrm{dc}}=0$), only the AC pump is unable to cause the incoherent gain for the probe field in the superlattice. Indeed, direct calculation [33,49] shows that in this case incoherent mobility
(12)$$\mu_{\mathrm{inc}}\left(E_{\mathrm{dc}}=0,E_{\mathrm{ac}}\right)=\mu_{0}{\left(1+F_{\mathrm{ac}}^{2}\right)}^{-3/2},\;F_{\mathrm{ac}}=E_{\mathrm{ac}}/E_{\mathrm{cr}},$$
is always positive. Based on a comparison of these two limiting cases and further analysis, one can conclude that within the incoherent gain mechanism, the power of the probe electric field is acquired predominantly from the DC source, while the AC pump only provides partial control over the process. Therefore, the incoherent gain is a sort of Bloch gain modified by the action of the AC pump field [38].

Explicit, but rather cumbersome expressions of the mobility components (Equation (Equation 9)) can be found in [33,36]. However, our focus here is solely on the numerical analysis of some characteristic cases important for our extended large-signal analysis (Figure 1). In this figure, the incoherent (Figure 1 left column), coherent (Figure 1 middle column) and maximum net gain (Figure 1 right column) are shown as functions of the scaled pump electric fields ($F_{\mathrm{dc}},F_{\mathrm{ac}}$) for three frequency ratios $\omega_{1}/\omega_{0}=$3/2, 1/2, 5/3. The locations of the net gain areas $\mu (\phi_{\mathrm{opt}})<0$ in the $F_{\mathrm{dc}}-F_{\mathrm{ac}}$ plane essentially determine the pump field conditions, providing the microwave signal generation (P<0). Whereas interplay between the coherent and incoherent gain components define the physical mechanisms of this process. Since the $\mu_{\mathrm{inc}}$ component does not depend on the frequency ratios, the incoherent gain area (colour) is the same for all three frequency cases, and it can be qualitatively described as satisfying $F_{\mathrm{dc}}>F_{\mathrm{ac}}$ (hereafter named as “type I biasing conditions”). In contrast, $\mu_{\mathrm{inc}}>0$ (blank) when the pump field strength is roughly such that $F_{\mathrm{dc}}<F_{\mathrm{ac}}$ (“type II biasing conditions”), and no incoherent gain exist. The coherent component of mobility is always non-zero ($\mu_{\mathrm{coh}}\left(\phi \right)\neq 0$) for various half-integer harmonics $n_{0}\omega_{0}/2$. Therefore, when the relative phase ϕ is chosen close to its optimal value $\phi =\phi_{\mathrm{opt}}$ the coherent gain arises ($\mu_{\mathrm{coh}}<0$), and its area occupies the whole plane (see the upper and middle subplots in the middle column of Figure 1). However, as the colour intensities in these subplots indicate, the locations and magnitudes of the coherent gain maxima are visibly different for 1/2- and 3/2-harmonics. This results in a significant structural difference of the net gain areas $\mu (\phi_{\mathrm{opt}})<0$ in these two characteristic frequency cases.

For the case of $3/2\;\omega_{0}$, the net mobility $\mu (\phi_{\mathrm{opt}})$ (Equation (Equation 8)) is negative in two well-separated generation regions. Namely, for the type II biasing, the coherent gain exceeds the incoherent absorption ($|\mu_{\mathrm{coh}}|>\mu_{\mathrm{inc}}$), while for type I biasing, both coherent and incoherent gain mechanisms coexist ($\mu_{\mathrm{coh}}<0$, $\mu_{\mathrm{inc}}<0$). The area of this second generation region, characterized by an interplay between two types of gain, practically coincides with the area of incoherent gain. We also found that the structure of several well-separated generation regions in the $F_{\mathrm{dc}}$-$F_{\mathrm{ac}}$ plane is typical for all high-order half-harmonics ($n_{0}=5,7,\ldots$). In contrast, the $1/2\;\omega_{0}$ case is characterized by a unique single generation region. However, there also exists a boundary separating the area of pure coherent gain and the interplay region of the two gain mechanisms (see the middle figure in the right column of Figure 1). A starkly different situation arises in the case of fractional frequencies (m/n,n≥3), exemplified with $5/3\;\omega_{0}$ in the bottom row of Figure 1. Since in this case $\mu_{\mathrm{coh}}=0$ for any pump field parameters, the possibility of generation at a fractional harmonics is solely connected to the incoherent gain $\mu =\mu_{\mathrm{inc}}<0$.

As argued earlier, the coherent and incoherent gain components in the DC–AC-driven superlattice can be viewed as the parametric gain and Bloch gain, respectively. The links between the phase-dependency of the parametric and phase-independency of the Bloch mechanisms are especially important and convenient since they remain valid beyond the small-signal approximation; however, the separation of *P* into the coherent and incoherent components cannot be mathematically performed.

We now turn our attention to the analysis of the large-signal gain, first needing to underline the importance of the relative phase (phase difference between the AC pump and probe fields), since distinct dependencies of the probe field on the phase may be recorded while changing the ratio $\omega_{1}/\omega_{0}$ and pump field parameters ($F_{\mathrm{dc}},F_{\mathrm{ac}}$). Several of such typical dependencies are depicted in Figure 2, where the coloured areas indicate positive gain (P<0) and blank areas mean no amplification. These calculations were performed using the integral in Equation (Equation 6). Figure 2A presents a pure parametric generation of the second harmonic in an unbiased superlattice, which is the simplest case in the class of harmonic and half-harmonic frequency generation. This gain diagram clearly shows well-defined optimal values of the relative phase (Equation (Equation 5)) that do not depend on the probe field, as well as a range of phase values where amplification is not possible for any probe strength. Within this generation region, the signal can grow until the gain vanishes at $F_{1}^{\left(s\right)}\approx 3$. Note that the probe field strength providing the maximum generation power is visibly less than the corresponding amplitude of steady-state oscillations, $F_{1}^{\left(s\right)}$. Figure 2B,C depicts the formation of amplification islands, typical for large-signal parametric processes involving fractional frequencies (*m/n*, with n≥3). In this situation, parametric generation evolving from the small signal is not available for any relative phase unless some mechanism, external with respect to the large-signal parametric mechanism (red arrows), is applied at this frequency to ignite the phase-dependent amplification process, further progressing within the island (green arrows). This simple concept works both with and without DC bias, appearing to be quite useful in explaining the fractional harmonic generation. In what follows, we consider several routes to reach the amplification islands, including the phase-independent mechanism of DC–AC–Bloch gain. Bearing in mind the experimental findings [28], we now proceed to conduct a detailed study of the high-frequency gain at various sub-harmonics in DC–AC-biased superlattices.

### 3.2. Pure Parametric Gain

For sub-harmonics there exists a diverse set of gain diagrams in the probe-phase plane, which is controlled not only by the pump field strengths $\left(E_{\mathrm{dc}},E_{\mathrm{ac}}\right)$ and frequency ratio $\omega_{1}/\omega_{0}$, but also by the amplitude of the probe field itself. Nevertheless, only two types of the net gain mechanisms contribute to this diversity: (i) The pure parametric gain, when the parametric gain overcomes the Bloch absorption and; (ii) a hybrid process involving a combination of the phase-independent Bloch and phase-dependent parametric gain, characterized by the interplay and competition between the two components.

We begin with the analysis of positive gain at high-order half-integer harmonics ($n_{0}\geq 3$). The pure parametric generation exists when the DC pump electric field is low, whilst the AC pump electric field is high (type II biasing), as depicted in Figure 3A for the case $3/2\;\omega_{0}$. As evident from the top inset, the phase dependence is peaking at optimal values π/2 and 3π/2 in Figure 3a,b, and additionally a large-signal splitting of the optimal phase is seen in Figure 3c. Such a splitting, however, is not typical for other gain diagrams obtained within the quasi-static approximation. Next, one can note the increase in the maximum probe electric field strength and the generated power with an increase in the AC pump field. In the case of Figure 3c, corresponding to the pump strengths $\left(F_{\mathrm{dc}},F_{\mathrm{ac}}\right)=\left(0.5,4\right)$, we estimate the maximum generated power of 3.2 mW at $E_{1}\approx 9.7$ kV/cm.

The considered generation process under type II biasing was essentially induced by the coherent component of the small-signal gain. The situation becomes different in the case of various fractional harmonics, for which this small-signal and phase-dependent gain are absent (cf. Figure 1), and therefore no signal generation from small fluctuations is allowed. Figure 3B illustrates the properties of the pure parametric amplification at fractional frequencies, in particular for $5/3\;\omega_{0}$. The gain regions take the form of the large-signal amplification islands (insets (d–f) of Figure 3), which have the periodicity of 2π/n, *n* being an odd denominator (cf. Figure 2B,C). Such an amplification at a specific fractional frequency can only begin if the probe field strength overcomes a characteristic field determining the lower boundary of the island. For example, in the case of the pump strengths ($F_{\mathrm{dc}},F_{\mathrm{ac}})=\left(0.5,4\right)$ represented in Figure 3d, the amplification begins at $F_{1}\approx 0.75$ and extends up to $F_{1}^{\left(s\right)}\approx 2$. Within this island a maximum power up to 3.3 mW can be reached at the signal strength $E_{1}\approx 8.8$ kV/cm, comparable with the power level earlier estimated for the pure parametric generation of $3/2\;\omega_{0}$. For fractional frequencies with an even denominator, the tendency of the island formation and its periodicity remain; however, the effect arises at significantly higher AC pump field strengths. For example, in the case of $5/4\;\omega_{0}$ the minimum AC pump field, required for a formation of the islands, is already $F_{\mathrm{ac}}\approx 8$. Naturally, the probe field amplitude, corresponding to the lower boundary of the island, increases when increasing the denominator of the amplified fractional frequency (cf. Figure 2B,C).

### 3.3. Coexistence of Bloch and Parametric Gain

Under type I biasing conditions, the Bloch gain can play a leading role in both half-harmonic and fractional frequency generation. However, this specific case of the pump field strength was not addressed in earlier key works [31,37] due to the criterion for the absence of electrical domains used in these papers. Namely, the papers relied on a variant of the stabilization method known as the limited space-charge accumulation mode [50,51], for which the necessary condition for the absence of electrical domains requires that the small-signal differential mobility (Equation (Equation 9)) is positive, that is $\mu_{\mathrm{inc}}>0$[31,52]. Obviously, this is in opposition to the inequality $\mu_{\mathrm{inc}}<0$ that must be satisfied following the definition of type I biasing. Within the NL-criterion utilized in this present work, there are no formal limitations on the sign of $\mu_{\mathrm{inc}}$, and therefore effects of the Bloch gain can be considered in detail as an important part of the realistic model analysis.

To begin with, we analyse the coexistence and competition of the parametric and Bloch mechanisms of large-signal gain for 1/2—harmonic generation. The gain diagrams for $1/2\;\omega_{0}$ display several unique features, thus giving the possibility to clearly distinguish the transition from the canonical Bloch gain ($F_{\mathrm{ac}}=0$) to the purely parametric process with an increase in AC pump amplitude (Figure 4). On the one hand, it is known that a large coherent component of the small-signal gain is an inherent feature of the 1/2—harmonic case (Figure 1), and the magnitude of this phase-dependent gain can further increase with an increase in the pump field. On the other hand, Bloch generation can exist without an AC pump (see Equation (Equation 11) and Figure 4a). Thus, for moderate AC pump strengths, the Bloch and parametric mechanisms begin to compete within the entire range of probe field amplitudes. With a further increase in the AC pump, the role of the Bloch generation diminishes. This is seen due to ever-increasing relative phase dependency on the generated probe electric field power (cf. (a–f) in the inset of Figure 4). Finally, the contribution of the Bloch generation process becomes low enough to be neglected, resulting in the complete dominance of the pure parametric gain (Figure 4f). Therefore, the transition from the phase-independent Bloch generation to the parametric one can be tuned just by sweeping the AC pump field strength.

In the case of type I biasing, the generated power is considerably larger compared to the pure parametric generation achieved in type II biasing conditions. The most powerful output is naturally expected in the $1/2\;\omega_{0}$ case. For example, the use of the pump field strengths $\left(F_{\mathrm{dc}},F_{\mathrm{ac}}\right)=\left(4,2\right)$ corresponding to Figure 4d, results in a power of 257 mW with a maximum generated signal of 31.5 kV/cm ($F_{1}^{\left(s\right)}\approx 5.75$).

It is worth noting that an additional appealing feature of the net gain at the $1/2\;\omega_{0}$ harmonic is its anomalously low threshold for the AC pump strength required for pure parametric generation of this sub-harmonic. Indeed, this threshold can be as small as $10^{-3}$ in terms of the critical field $E_{\mathrm{cr}}$ (Figure 4B). In other words, the AC pump field only requires providing seeding to reach the gain at $1/2\;\omega_{0}$; hence, this behaviour somewhat resembles the seed-related effects known in optically pumped parametric oscillators. This is in sharp contrast with the 3/2—harmonic case, where the AC pump electric field strength must overcome a significant threshold ($F_{\mathrm{ac}}\geq 1.8$) for generation (Figure 3B). Yet, another interesting feature of the net gain behaviour in $1/2\;\omega_{0}$ frequency case—the possibility of switching from parametric to predominantly Bloch generation by sweeping the DC bias and simultaneously keeping the AC pump negligibly small. The effect is also only inherent for the $1/2\;\omega_{0}$ frequency, as depicted in Figure 4B. One can note that the transition interval of the pump DC electric field is very small (approx. $0.1E_{\mathrm{cr}}$), thus revealing an additional simple and relatively small power-required switching possibility between the two different generation effects.

We now proceed to fractional harmonic generation under type I biasing conditions, disclosing a surprising transformation of the completely phase-independent gain to the predominantly phase-dependent amplification with growth of the generated signal. This transformation, also seen as an effective conversion, is exemplified in Figure 5 where the dependence of the probe electric field strength on the relative phase of the fractional $5/3\;\omega_{0}$ (upper row) and $5/4\;\omega_{0}$ (bottom row) frequencies is highlighted. The calculations are performed using the same values of the pump electric fields as in the $1/2\;\omega_{0}$ case given in the upper inset of Figure 4. For type I biasing (marked by the solid red line in the main graphs of Figure 1 and Figure 3A,B) and fractional frequencies, the incoherent component of the small-signal mobility is negative $\left(\mu_{\mathrm{inc}}<0\right)$ but the corresponding coherent component approaches zero. This ensures a phase-insensitive initiation of the generation process throughout the whole bias locus. However, as evident from Figure 5, phase dependence emerges at larger probe field amplitudes, thus implying the somewhat paradoxical existence of the parametric gain under type I biasing conditions. To explain this transformation to the parametric gain type at moderate probe field amplitudes, we explore the concept of amplification islands corresponding to fractional frequencies. Indeed, if the AC pump field is strong enough, a reasonably sized amplification island with a parametric nature is formed (Figure 2 and Figure 3). Since the Bloch gain typically saturates at large signal strengths [38,40,52], the Bloch and parametric processes can overlap and become hybridized within the amplification island, resulting in the appearance of phase-dependency of the net gain. This simple picture, despite being instructive, is incomplete as the hybrid generation mode is more complex than a direct overlap of two uncombined processes; in particular, the hybrid mode can saturate at a different probe field amplitude from the saturation amplitude of Bloch gain itself.

We estimated the generated power at the $5/3\;\omega_{0}$ by employing the same pump strengths as those used earlier for $1/2\;\omega_{0}$ ($\left(F_{\mathrm{dc}},F_{\mathrm{ac}}\right)=\left(4,2\right)$ in Figure 5d). In this case the maximum power and generated signal of 43.2 mW and 20 kV/cm ($F_{1}^{\left(s\right)}\approx 3.65$) was found, respectively. As expected, the power generated at fractional frequencies within the hybrid mode is significantly lower compared to the 1/2-harmonic; however, it is still an order of magnitude larger compared to the power delivered at the same fractional frequency in modes of pure parametric generation and island-related amplification.

Based on further numerical simulations involving different fractions m/n, we arrived at the conclusion that the described hybrid gain mode is a universal tool to reach the large-signal and predominantly phase-dependent generation of various fractional harmonics. Therefore, a combination of the degenerate Bloch–parametric mechanism considered here, along with the large-signal mechanism of the non-degenerate multi-photon processes (Equation (Equation 15), Appendix A), provides a reliable qualitative explanation for the physical origin of the mutual modes with fractional frequencies observed here [28].

### 3.4. Pure Parametric Fractional Frequency Generation via Ignition

In terms of generation conditions, it is worth noting that the ignition in the region of pure generation islands (type II biasing) can be induced by the external electric field of the corresponding fractional frequency (Figure 2). Alternatively, the ignition can also be achieved by changing the primary pumping conditions; for instance, switching the DC pump electric field from an “on” state, when the applied electric fields satisfy the generation conditions from small signals, to an “off” state, representing an amplification island with a possible amplification initiation from some boundary probe field. For a particular case, covering the $5/3\;\omega_{0}$ frequency, the “on” state corresponds to the $\left(F_{\mathrm{dc}},F_{\mathrm{ac}}\right)=\left(4.7,4\right)$ pump, while the “off” state corresponds to the $\left(F_{\mathrm{dc}},F_{\mathrm{ac}}\right)=\left(0,4\right)$ pump (Figure 6A). A deeper insight can be obtained from Figure 6B, where the dependencies of the probe electric field on the relative phase in the aforementioned “on” and “off” states are presented. As one can see, the pictures are fundamentally different. The optimal and constant phase are selected according to the amplification islands in the “off” state, resulting in a non-optimal phase in the “on” state due to π/3 optimal phase shift. The physics behind these features are the above-described strong interplay of the Bloch gain, incoherent in its origin, and the phase-defined parametric gain processes. In the “off” state, when the DC electric field $F_{\mathrm{dc}}=0$, the phase role begins to predominate, inducing strong expression of the prevailing pure parametric gain effect. We observed that in this “off” state the amplification island was induced when the probe electric field exceeded $F_{1}>0.5$, slightly below the probe field strength achieved during the “on” state generation process. The amplification island extends up to $F_{1}^{\left(s\right)}=2.5$, marking the maximum possible probe field strength reached when employing switching ignition for $F_{\mathrm{ac}}=4$.

## 4. Concluding Remarks and Outlook

Motivated by recent experiments and their interpretation within the small-signal model [28], we conducted a theoretical study on the peculiarities of sub-harmonic generation in miniband superlattices driven by a combination of DC and AC electric fields. In contrast to the earlier works, we paid special attention to the conditions when parametric [31] and Bloch gain [38,40] can coexist, also going beyond the small-signal approximation considering a number of novel large-signal effects. Using quasi-static approximation, valid for GHz–sub-THz frequencies, we investigated conditions for positive high-frequency gain in major types of half-integer and fractional harmonics of the AC pump electric field. Among our findings the most intriguing is the examination of the conversion of phase-independent Bloch gain to the phase-dependent parametric processes in a relatively strongly DC-biased superlattice. This surprisingly robust large-signal hybrid process of the fractional frequency generation is in line with corresponding experimental observations. In wide-miniband superlattices with modest doping, our calculations predict application-attractive generation powers reaching hundreds of mW for half-harmonics. Additionally, we established a method for fractional frequency generation by ignition of the pure parametric gain mechanism. This method relies on use of only DC bias switching, contributing to the ever-expanding pool of gain ignition techniques known for semiconductor superlattices [38] and quantum cascade lasers [53].

To summarize, this study exposed a ground-breaking new avenue for high-frequency signal generation and amplification in semiconductor nanostructures at room temperature. As the lack of compact powerful sources in the THz range is one of the most challenging issues in promising new THz applications, the proposed approach to employ semiconducting superlattices for THz amplification is important in stimulating the evolution of compact solid-state-based sources within broadband GHz–THz ranges. This could significantly impact the development of THz technology [20] and, in particular, accelerate the improvement and wider implementations of compact spectroscopic THz imaging systems [54] under real operational conditions. The other attractive pathway relies on the ability of miniband superlattices to generate the output signal of a frequency as a sub-harmonic of the input signal. Thus, superlattices could potentially form the core of analogue frequency dividers. Currently, microwave frequency dividers have many important applications in digital electronics, telecommunications, and metrology ([55] and references cited therein). Our findings demonstrate good prospect for the further development of new types of miniature frequency dividers operating in the GHz–sub-THz frequency domains.

## Figures and Tables

**Figure 1 nanomaterials-13-01993-f001:**
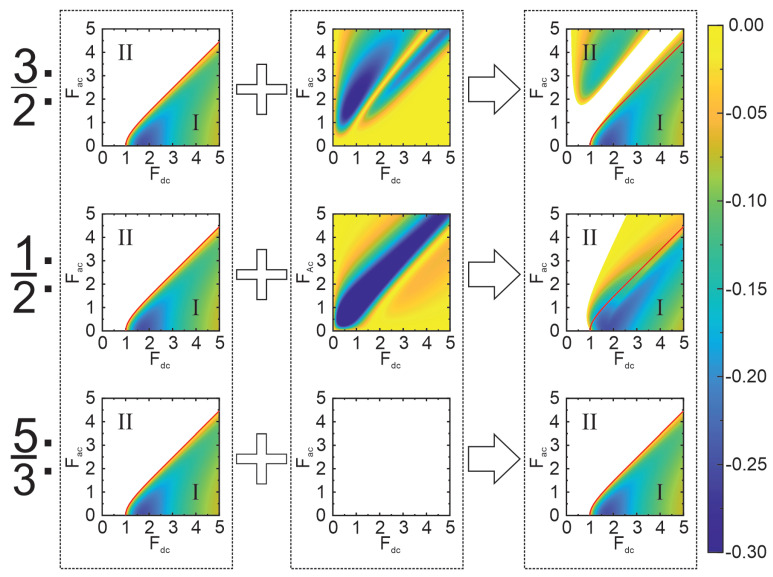
Dependencies of the small-signal mobilities (left—incoherent, centre—coherent, right—total) on the applied DC and AC pump field strengths for $3/2\;\omega_{0}$, $1/2\;\omega_{0}$, and $5/3\;\omega_{0}$ frequencies, calculated according to Equations (Equation 8) and (Equation 9). The coherent mobilities are calculated at the proper optimal values of the phase, and all mobilities are scaled to the Drude mobility $\mu_{0}$. The colours represent the negative values of the mobility components, displaying conditions for the gain associated with the corresponding mobility type. Blank areas mean positive or zero mobility values, thus displaying conditions when the generation is not possible within the small-signal approximation. Solid red lines outline the boundaries of the incoherent gain locus. Both the mathematical model and graphs clearly demonstrate that the incoherent component of the mobility is frequency independent, thus the differences in the total mobility depend only on differences in the coherent component. The total gain diagram of $3/2\;\omega_{0}$, which is representative for the generation of harmonics and half-harmonics, results in two separated generation areas, the upper left (*II*) being purely coherent and the bottom right (*I*) being a mixture of coherent and incoherent components. The case of $1/2\;\omega_{0}$ is interesting because a specific behaviour of the corresponding coherent component results in the total mobility being in a single area. The physical consequences of such dependence are discussed in Section 3.3. The case of $5/3\;\omega_{0}$ is typical for the generation of various fractional harmonics: the coherent component vanishes, thus the generation is caused solely by the incoherent gain only. These diagrams of the small-signal gain at characteristic frequencies provide a basis for our further detailed analysis of the large-signal amplification and generation.

**Figure 2 nanomaterials-13-01993-f002:**
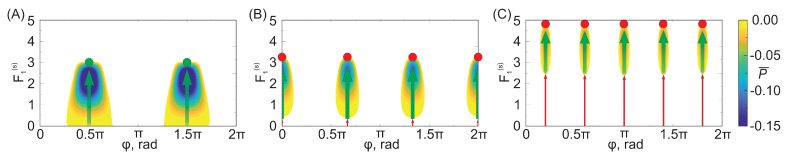
Exemplary regions of the large-signal gain ($\bar{P}<0$) shown in the plane of normalized signal amplitude $F_{1}$ and relative phase ϕ for several characteristic frequency ratios and fixed pump electric field components: (**A**) $\omega_{1}/\omega_{0}=2$, $F_{\mathrm{dc}}=0$, and $F_{\mathrm{ac}}=5$. The generation process is characterized by a positive gain both for small and large signals when the phase values are close to two green dots, marking the optimal phases $\phi_{\mathrm{opt}}=\pi /2$ and 3π/2. (**B**) $\omega_{1}/\omega_{0}=5/3$, $F_{\mathrm{dc}}=0$, and $F_{\mathrm{ac}}=5$. Large signal gain arises around the optimal phases (red dots, Equation (Equation 5)) only if the probe electric field strength exceeds a threshold. So-called *amplification islands* are formed. (**C**) $\omega_{1}/\omega_{0}=7/5$, $F_{\mathrm{dc}}=1$, and $F_{\mathrm{ac}}=10$. The probe electric field requires to overcome a large threshold to achieve gain. By comparing subplots (**B**) and (**C**) one can note the difference in the optimal phases and their periodicity. Everywhere the colour palette is used to indicate the power $\bar{P}=P/P_{0}$ that can be generated within the region of the gain at every given set (ϕ, $F_{1}$). Areas left blank, indicate no gain (P>0). Green arrows depict the way the signal can increase through the gain regions. Thin red arrows sketch how an additional ignition mechanism can bring the system into the large-signal gain regime.

**Figure 3 nanomaterials-13-01993-f003:**
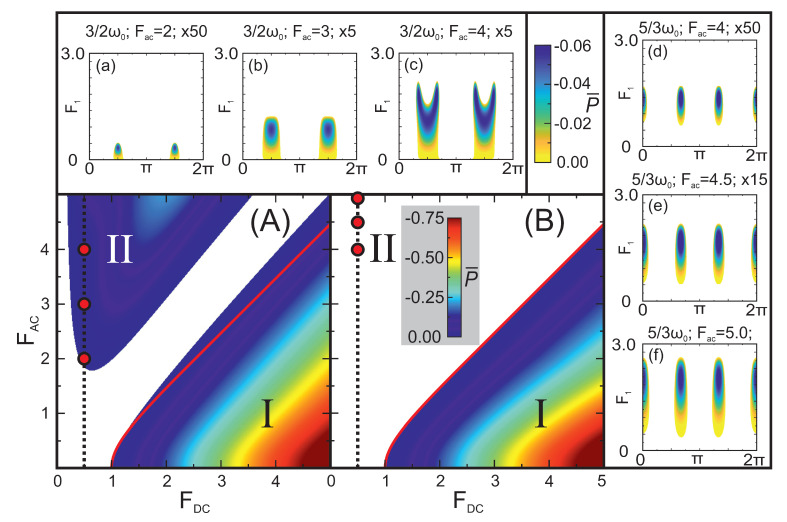
The pure parametric generation of half-harmonics and pure parametric amplification of fractional harmonics (large-signal amplification islands). Blank areas denote regions where generation is absent, while colours represent the power $\bar{P}$ value according to Equation (Equation 6). The red lines show the incoherent generation boundary of the small-signal model. (**A**) Maximum generation power dependency on the applied pump electric field (DC and AC) for $3/2\;\omega_{0}$. The black dashed line and red points, chosen from the type II biasing conditions, show the pump field strengths used to represent the signal vs. phase diagrams in the top inset (**a**–**c**)). Top inset (**a**–**c**): The signal field dependencies on the relative phase for the pump field strengths $F_{\mathrm{dc}}=0.5$ and $F_{\mathrm{ac}}=$ 2:1:4. We underline that the pure parametric generation at the 3/2-harmonic requires overcoming a significant threshold in AC pump strength ($F_{\mathrm{ac}}≳2$). (**B**): Maximum generation power dependencies on the applied pump electric field (DC and AC) for $5/3\;\omega_{0}$. The black dashed line and red points show the pump field strengths selected to depict the diagrams in the right inset (**d**–**f**). Inset (**d**–**f**): Probe electric field dependencies on the relative phase presented for the following pump parameters: $F_{\mathrm{dc}}=0.5$ and $F_{\mathrm{ac}}=$ 4:0.5:5. Note the formation of the large-signal amplification islands with well-defined lower and upper boundaries in the strength of the probe electric field.

**Figure 4 nanomaterials-13-01993-f004:**
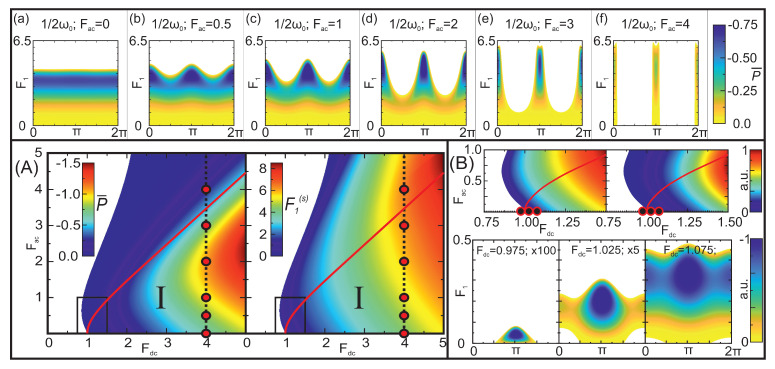
Coexistence of the parametric and Bloch gain mechanisms in the 1/2—harmonic generation. (**A**) Maximum power $\bar{P}$ (colour, left plot) and the maximum probe field strength (colour, right plot) generated at the optimal phase values. Red thick line marks the boundary of the small-signal incoherent gain ($\mu_{\mathrm{inc}}<0$, cf. Figure 1). Within the area below this line (region *I*), the incoherent and coherent gain mechanisms coexist and can compete. Six red dots on the vertical dashed line indicate the fixed pump electric field strengths ($F_{\mathrm{dc}}=4,F_{\mathrm{ac}}=$ 0:4) used to separately depict regions of the large-signal gain (P<0, colour) in the plane signal vs. phase. These six diagrams, located in the upper long panel and marked from (**a**) to (**f**), clearly expose the transition from completely phase-independent Bloch gain (diagram (**a**)) to strictly parametric generation (diagram (**f**)). (**B**) The upper half presents a zoomed-in view of the generation region framed in Figure 4A. This is located near the critical electric field $F_{\mathrm{dc}}=1$ and at the small AC pump electric field strengths $F_{\mathrm{ac}}<1$. Three black dots, corresponding to $F_{\mathrm{ac}}=10^{-3}$ and $F_{\mathrm{dc}}$ only slightly below and above the critical electric field, indicate the pump parameters chosen to depict the evolution of the large-signal generation regimes in three diagrams in the bottom half. These diagrams expose the interesting peculiarities of the transition from the pure parametric to dominantly Bloch generation at extremely low levels of the AC pump field.

**Figure 5 nanomaterials-13-01993-f005:**
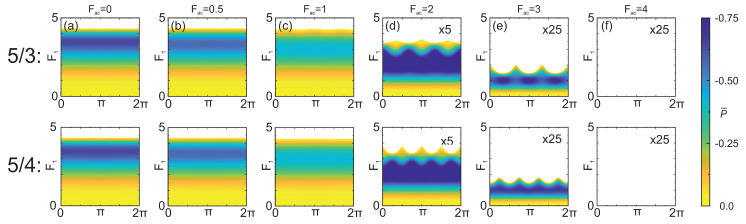
Transformation of the phase-independent Bloch gain to predominantly parametric gain in the hybrid process of fractional frequency generation. Gain diagrams, showing the dependencies of the probe electric field on the relative phase, are depicted for different pump conditions ($F_{\mathrm{dc}}=4$ and $F_{\mathrm{ac}}=$ 0:4, inset of Figure 4) for the fractional $5/3\;\omega_{0}$ (upper row) and $5/4\;\omega_{0}$ (bottom row) frequencies. Colours show the negative values of $\bar{P}$ calculated according to Equation (Equation 6), and blank areas indicate that $\bar{P}>0$ where no generation is possible. For a relatively small AC pump field $\left(F_{\mathrm{ac}}<2\right)$ the domination of the Bloch gain is expressed in the absence of phase dependency of the probe electric field. However, as the AC pump strength increases, phase-dependent amplification appears, suggesting the formation of large-signal amplification islands. Within these islands, the hybrid gain process exhibits a strong parametric component. Note the absence of generation when the AC pump field exceeds $F_{\mathrm{ac}}=4$ and no Bloch gain exists.

**Figure 6 nanomaterials-13-01993-f006:**
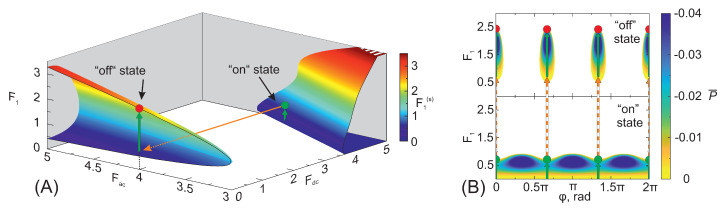
(**A**) Generated probe electric field profile dependency on the applied pump electric field (DC and AC) for $5/3\;\omega_{0}$, illustrating the ignition into the amplification island process via switching the DC pump electric field. The orange arrow represents the state switch from “on” $\left(F_{\mathrm{dc}},F_{\mathrm{ac}}\right)=\left(4.7,4\right)$ to “off” $\left(F_{\mathrm{dc}},F_{\mathrm{ac}}\right)=\left(0,4\right)$. Such a change of state allows the pure parametric generation to be reached in amplification islands from a negligibly small probe electric field without applying external AC ignition. (**B**): Dependence of the probe electric field $F_{1}$ on the relative phase, corresponding to the states given in Figure 6A (the bottom subplot—“on state” $F_{\mathrm{dc}}=4.7$; the upper subplot—“off state” $F_{\mathrm{dc}}=0$). Blank areas depict conditions where the generation or amplification is not possible. Green arrows represent amplification occurring inside the superlattice, while the dotted orange arrows represent the state change due to switching (cf. Figure 6A).

## Data Availability

The data presented in this study are available on request from the corresponding author.

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
