# Peer review of "Coexistence of Bloch and Parametric Mechanisms of High-Frequency Gain in Doped Superlattices"

_nanomaterials, 2023, doi:10.3390/nano13131993_

Round 1
Reviewer 1 Report
The manuscript Coexistence of Bloch and parametric mechanisms of high-frequency gain in doped semiconductor superlattices by Vladislovas Cižas, Natalia Alexeeva, Kirill N. Alekseev and Gintaras Valušis is a theoretical study of high-frequency signal gain in semiconductor superlattices, for the case of a probe microwave signal is comparable to the ac pump electric field.
The manuscript is generally clearly written although some review of English style is needed.
I suggest rewriting Introduction from line 75 to the end of the section. The information presented there, already contains parts more appropriate for discussion and conclusions.
From the section 2 Main equations and brief explanations is not clear what theoretical approach was used and to which kind of superlattices is can be applied. Term “semiconductor superlattices” is quite broad and imprecise. Which crystallographic structure does have a supperlattice?
The capture of Figure 1 should be more concise. The respective explanations have to be integrated in the main text. Also, it is not clear how the images were generated.
The same for all Figures in the manuscript. Such organization of the text makes the understanding difficult.
Conclusion part seems to be clear and well-written.
Moderate editing of English language style is required.
Reviewer 2 Report
In this manuscript, the authors have theoretically reported the coexistence of Bloch and parametric mechanisms of high-frequency gain in doped superlattices, which can be tuned by sweeping ac pump strength and leveraging a proper phase between the pump and strong probe electric fields, thereby providing a new path for miniature GHz-THz frequency generators, amplifiers, and dividers. After carefully reading the manuscript, my suggestion is minor revision. All the comments are listed as followed.
1. Some parts of the article don't read smoothly. Proper polishing of English is necessary.
2. In Introduction, the authors give five statements, which are obtrusive and useless. Therefore, I suggest that they can be deleted.
3. Similarly, in the section of Conclusions, one paragraph is enough for the conclusion. There is no need to write a lot of irrelevant stuff.
In this manuscript, the authors have theoretically reported the coexistence of Bloch and parametric mechanisms of high-frequency gain in doped superlattices, which can be tuned by sweeping ac pump strength and leveraging a proper phase between the pump and strong probe electric fields, thereby providing a new path for miniature GHz-THz frequency generators, amplifiers, and dividers. After carefully reading the manuscript, my suggestion is minor revision. All the comments are listed as followed.
1. Some parts of the article don't read smoothly. Proper polishing of English is necessary.
2. In Introduction, the authors give five statements, which are obtrusive and useless. Therefore, I suggest that they can be deleted.
3. Similarly, in the section of Conclusions, one paragraph is enough for the conclusion. There is no need to write a lot of irrelevant stuff.
Reviewer 3 Report
The manuscript by Cizas et al. entitled “"Coexistence of Bloch and Parametric Mechanisms of High-Frequency Gain in Doped Superlattices" is investigates high frequency gain in doped superlattices by theoretical considerations. The results can be interesting for developing novel miniature GHz-THz generators, amplifiers, and dividers for room temperature operation which could be exciting for the physicists as well as material scientists. However, some issues need to be rectified before publication.
1. The introduction should be improved with recent developments in this area. What is the originality of this work as compared to the existing investigations in this field?
2. The cited references are two old for example, 1970, 1972, etc. The authors should introduce recent references to increase the innovation of this work.
3. Most of the equations 1-3, are well known. What is the highlight here?
4. The caption of Fig. 1 is unnecessarily too large. The authors should consider it dividing and merging it to the main text for better readability.
5. What is Figure 1 middle column? The authors should improve the presentation of this Figure.
6. Similarly, for Figure 5. Why don’t you divide the all small figures into different sub captions?
7. Please highlight the captions A and B in Figure 6. It is hardly visible.
8. The font size of images are too small. The authors should improve the font size of the images.
9. The major findings should be highlighted in bullets in conclusion section.
10. Additionally, English should be improved.
There are many large sentence structures in introduction which should be avoided for better readability.
Round 2
Reviewer 1 Report
The manuscript can be published now.
English language is generally fine, some minor editing could improve the manuscript quality.